# Investigation of Hybrid Films Based on Fluorinated Silica Materials Prepared by Sol–Gel Processing

Violeta Purcar [1], Valentin Rădiţoiu [1], Florentina Monica Raduly [1], Alina Răditoiu [1,*], Simona Căprărescu [2], Adriana Nicoleta Frone [1], Cristian-Andi Nicolae [1] and Mihai Anastasescu [3]

[1] National Institute for Research & Development in Chemistry and Petrochemistry—ICECHIM, Splaiul Independentei no. 202, 6th District, 060021 Bucharest, Romania; violeta.purcar@icechim.ro (V.P.); vraditoiu@icechim.ro (V.R.); monica.raduly@icechim.ro (F.M.R.); ciucu_adriana@yahoo.com (A.N.F.); ca_nicolae@yahoo.com (C.-A.N.)

[2] Faculty of Chemistry Engineering and Biotechnologies, Department of Inorganic Chemistry, Physical Chemistry and Electrochemistry, University Politehnica of Bucharest, Ghe. Polizu Street, no. 1–7, 6th District, 011061 Bucharest, Romania; simona.caprarescu@upb.ro

[3] Institute of Physical Chemistry "Ilie Murgulescu" of the Romanian Academy, Splaiul Independentei no. 202, 6th District, 060021 Bucharest, Romania; manastasescu_ro@yahoo.com

* Correspondence: coloranti@icechim.ro

**Abstract:** In this research, fluorinated silica materials were prepared through sol–gel processing with tetraethylorthosilicate (TEOS), triethoxymethylsilane (MTES), and trimethoxyhexadecylsilane (HDTMES), using a fluorinated solution (FS) under acidic medium. The fluorinated solution (FS) was obtained by diluting the perfluorooctanoic acid (PFOA) in 2-propanol. These fluorinated sol–gel silica materials were placed on the glass surfaces in order to achieve the antireflective and hydrophobic fluorinated hybrid films. The structure and surface properties of the final samples were investigated by Fourier transform infrared spectroscopy (FTIR), ultraviolet/visible spectroscopy, thermogravimetric analysis (TGA), atomic force microscopy (AFM), and contact angle (CA) determinations. FTIR spectra demonstrated the presence of a silica network modified with alkyl and fluoroalkyl groups. Thermal analysis showed that the fluorinated sol–gel silica materials prepared with HDTMES have a good thermostability in comparison with other samples. Ultraviolet/visible spectra indicated that the fluorinated hybrid films present a reflectance of ~9.5%, measured at 550 nm. The water contact angle analysis found that the wettability of fluorinated hybrid films was changed from hydrophilic (64°) to hydrophobic (~104°). These hybrid films based on fluorinated sol–gel silica materials can be useful in various electronics and optics fields.

**Keywords:** sol–gel processing; fluorinated silica materials; hybrid films; wettability

## 1. Introduction

In recent years, it has been observed that the fluorinated materials obtained by the incorporation of fluorinated compounds in functional materials (e.g., blended composites, hydrogels, and functionalized surfaces), have been used in various applications, such as imaging [1], electronics/optoelectronics [2,3], conductors [4,5], surface coatings [6,7], and the automotive industry [8]. Different functional silica materials were prepared by various methods such as the sol–gel method [9], centrifugal casting [10], liquid-phase deposition [11], and spin-coating [12]. The sol–gel method has been widely used to fabricate organic–inorganic hybrid materials based on fluorinated compounds, and hydrophobic organic thin films due to their advantages such as simple processing, controllability, versatility, feasibility, low-temperature processing, lower maintenance, and repeatability.

Various silicone and fluoroalkyl-modified materials have been used as agents for the modification of surfaces in the different fields of coatings, films, adhesives, and fibers,

due to their hydrophobic/super-hydrophobic properties [13–15]. Oldani et al. [13] indicated that the sol–gel method was successfully used for the synthesis of silica combined with tetraethyl orthosilicate and a perfluoropolyether. The obtained mix was used on a steel surface. The results indicated that the obtained materials adhere to the surface of the steel, and have significant properties, such as hydrophobic character, anti-fouling, and good resistance to chemical demands in liquid environments. Ke et al. [15] used the sol–gel method to achieve organic–inorganic materials based on silica which were used to coat glass surfaces. They concluded that the coatings have a superhydrophobic character. This observation was reported due to the obtained values of the contact angle of 154°, after the decoration process with fluorosilane.

In the previous research, it was reported that the different hybrid-modified materials can be prepared by controlling different parameters (e.g., temperature, amount of nanoparticle, and stirring speed) [16,17]. It was demonstrated that the surface of the prepared silica ($SiO_2$)/polystyrene (PS) nanocomposites could be changed from superhydrophilic to superhydrophobic by controlling the amount of $SiO_2$ nanoparticles and the drying temperature [16]. Startek et al. [17] indicated that the sol–gel method was successfully useful to prepare hybrid materials based on modified and fluorinated silica nanoparticles. They demonstrated that the silica network observed in the hybrid materials can be changed using organically modified silanes containing fluoroalkyl chains. Yu and Xu [18] realized fluorinated hybrid films using the sol–gel method. Their study demonstrated that the obtained organic–inorganic films had fluorinated side chains derived from tridecafluoroctyltriethoxysilane and the silica network. The obtained hybrid films with low surface energy could be utilized as functional water-repellent coatings. Maehana et al. [19] reported that silica loaded with fluorine can be achieved through the sol–gel method, using hydrofluoric acid. They showed that the presented study is useful in obtaining and developing different optical materials. Uğur et al. [20] prepared non-stick fluorine-containing ceramic coatings through sol–gel processing for aluminum surfaces. The results showed that the coatings present good adhesion on substrates and hydrophobic properties (contact angle of about 106°). Rivero et al. [21] demonstrated that the hybrid matrix, obtained by the co-polymerization of organically modified silica alkoxides and modified with a silica precursor based on fluorinated polymeric chains, can improve the properties of an aluminum alloy. The resulting coatings presented good mechanical durability and hydrophobic property (contact angle of ~121°). Rosace et al. [22] prepared stable hydrophobic, non-toxic, anti-fouling coatings, using hybrid organic–inorganic materials obtained by the co-condensation of silane crosslinking agents with epoxide and amine tail groups, bended with different perfluorosilane precursors. In our previous studies, we have successfully demonstrated that hydrophobic and antireflective films can be achieved by sol–gel processing, using compositions prepared with perfluoroalkylsilane (FAS13) and tetraethylorthosilicate (TEOS) at different molar ratios [23]. It has been shown that hydrophobicity and optical transparency can be accomplished simultaneously by controlling the surface roughness.

The preparation and characterization of fluorinated sol–gel silica materials are presented in this work. These materials were obtained by using silane precursors with different lengths of alkyl chains (tetraethylorthosilicate (TEOS), triethoxymethylsilane (MTES), trimethoxyhexadecylsilane (HDTMES)), and a fluorinated solution (FS) obtained by diluting perfluorooctanoic acid (PFOA) in 2-propanol. The sol–gel reactions were performed under an acidic medium (2-propanol + HCl 0.1 N), at a temperature of 60 °C ± 2 °C. The series of fluorinated hybrid films were achieved through the deposition of fluorinated sol–gel silica materials on glass surfaces. The structure and surface properties of the final samples were tested by FTIR-ATR and ultraviolet/visible spectroscopy, TGA analysis, AFM, and water contact angle determinations.

## 2. Reagents and Apparatus

### 2.1. Reagents

Tetraethylorthosilicate (($C_2H_5O)_4Si$, TEOS, 98% purity, as silica source), triethoxymethylsilane ($C_7H_{18}O_3Si$, MTES, 99% purity, as modifier silane agent), and trimethoxyhexadecylsilane ($C_{19}H_{42}O_3Si$, HDTMES, 85% purity, as modifier silane agent) were purchased from Aldrich (Saint Louis, MO, USA). Perfluorooctanoic acid ($C_8HF_{15}O_2$, PFOA, 95% purity, as a fluorinated silicate) was bought from Aldrich (Saint Louis, MO, USA). Aluminum isopropoxide ($C_9H_{21}O_3Al$, AIP, 98% purity, as a reaction catalyst), maleic acid anhydride ($C_4H_2O_3$, MA, 99.7% purity, as a complexing agent), 2-propanol (99.9% purity, as a solvent) and hydrochloric acid solution (HCl 0.1 N, as an acid catalyst) were bought from Sigma-Aldrich Co. (Saint Louis, MO, USA). The materials and solvents were used as received, without any further purification.

### 2.2. Preparation of Fluorinated Sol–gel Silica Materials and Fluorinated Hybrid Films

Fluorinated sol–gel silica materials were prepared via the sol–gel route, under an acidic medium (using 2-propanol and HCl 0.1 N) at a temperature of 60 °C ± 2 °C, adding different silane precursors (TEOS, MTES, HDTMES) and a fluorinated solution (FS).

For the preparation of fluorinated sol–gel silica materials, the synthesis procedure was as follows: 2-propanol (10 mL), hydrochloric acid solution (HCl 0.1 N, 0.4 mL), TEOS, MTES, and/or HDTMES, and fluorinated solution (FS) were mixed together and stirred magnetically for 1 h at a temperature of 60 °C ± 2 °C. At this stage, the silane precursors were added in the molar ratio of TEOS:MTES = 1:1 and of MTES:HDTMES = 1:0.2. Afterward, MA (0.04 g), aluminum isopropoxide solution (AIP solution, 0.2 mL), and HCl 0.1 N (1 mL) were added into the mixture and then reacted for another 1 h. Because aluminum isopropoxide is very reactive, the addition of maleic anhydride as a complexing agent is necessary to control both the hydrolysis and condensation rate. Alkoxysilyl groups can actually co-condense for the period of the condensation reaction implicated in the sol–gel processing, thus conducting the generation of the interaction between organic and inorganic groups. In acidic conditions, the rate of hydrolysis is higher than the rate of condensation of the obtained materials.

The fluorinated solution (FS) was obtained by diluting the perfluorooctanoic acid (PFOA, 0.02 g) in 2-propanol (10 mL) at a temperature of 60 °C ± 2 °C.

The aluminum isopropoxide solution (AIP solution) was prepared by dissolving aluminum isopropoxide (AIP, 0.02 g) in 2-propanol (10 mL) at room temperature (25 °C ± 2 °C). A small amount (0.2 mL) from the AIP solution was used to prepare the fluorinated sol–gel silica materials.

Finally, four fluorinated sol–gel silica materials (stable and homogeneous) were obtained and notated as: TEOS+FS (S1), TEOS+MTES+FS (S2), TEOS+HDTMES+FS (S3), and TEOS+MTES+HDTMES+FS (S4). These final samples were characterized as white powders (sited in plastic vials, dried, and milled) and as fluorinated hybrid films (placed on glass surfaces (one side only)).

The glass surfaces, before use, were washed with the soap solution for 20 min in an ultrasonic bath. After that, the glass surfaces were cleaned with acetone, ethyl alcohol, and deionized water. The steps of washing and cleaning were repeated three times. Then, the glass surfaces were introduced in the desiccator and dried for 24 h under a vacuum to clean away the probable residues and chemical impurities, and to obtain a high quality of deposited films.

The schematic diagram of the realization process of the fluorinated sol–gel silica material and the fluorinated hybrid film is shown in Figure 1.

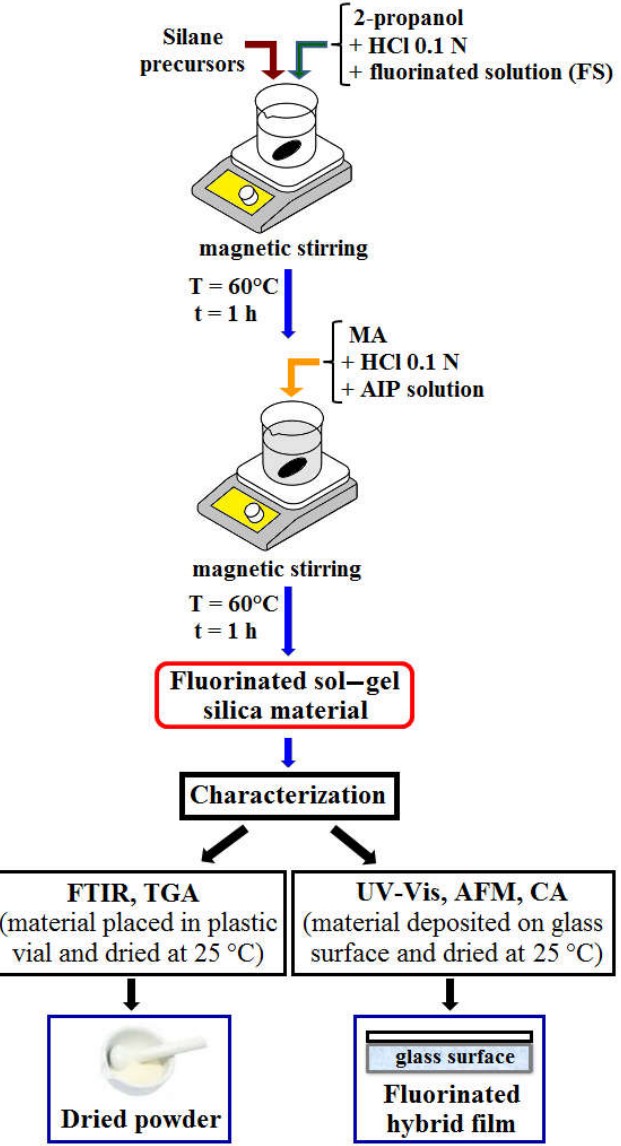

**Figure 1.** Schematic diagram of the realization process of the fluorinated sol–gel silica material and the fluorinated hybrid film.

### 2.3. Characterization of the Sol–gel Fluorinated Silica Materials and Hybrid Films

2.3.1. Fourier Transform Infrared Spectroscopy (FTIR)

The infrared spectra of the fluorinated sol–gel silica materials (as powders) were recorded on a Jasco FT-IR 6300 instrument (JASCO Int. Co., Ltd., Cremella, Italy), provided with the attenuated total reflectance (ATR) diamond attachment. FTIR spectra were investigated in the 4000 to 400 cm$^{-1}$ range (resolution of 4 cm$^{-1}$, 30 scans) at room temperature.

2.3.2. Thermogravimetric Analysis (TGA)

The thermal properties of fluorinated sol–gel silica materials (as powders) were analyzed using a TGA Q500 instrument (TA Instruments, Eschborn, Germany) under a nitrogen atmosphere. The samples (9–11 mg) were placed in aluminum pans and heated from 30 to 750 °C with a 10 °C/min rate (three TGA runs were recorded for each sample).

### 2.3.3. Ultraviolet/Visible Spectroscopy

The optical properties of fluorinated hybrid films were investigated through the UV-VIS-NIR-Jasco V-570 spectrophotometer equipped with a large integrating sphere ILN-472 (JASCO Int. Co., Ltd., Cremella, Italy). Transmittance and reflectance were analyzed in the range of 380–780 nm. Each sample was measured three times (standard deviation ± 0.2%).

### 2.3.4. Atomic Force Microscopy (AFM) Measurements

Atomic force microscopy (AFM) analysis was measured in non-contact mode with an XE-100 (Park Systems), which is equipped with flexure-guided, cross-talk eliminated scanners. The AFM images were rendered with sharp tips (NSC from NanosensorsTM) with ~8 nm radius of curvature, 90 μm mean length, 32 μm mean width, 2 N/m force constant, and 130 kHz resonance frequency. The XEI program (v 1.8.0—Park Systems) was used for the processing of the AFM images to evaluate the roughness of the fluorinated hybrid films.

### 2.3.5. Contact Angle (CA) Measurements

The wettability properties of fluorinated hybrid films were tested in a static regime at an ambient temperature, using a Tensiometer (KSV Instrument CAM 200, Helsinki, Finland) coupled with a high-resolution camera (Basler A602f, Ahrensburg, Germany). The values of the contact angles were identified with a water drop volume of about 6 μL. Water droplets were placed carefully in different regions of the fluorinated hybrid films (average of ten single determinations for each sample).

## 3. Results

### *3.1. FTIR-ATR Spectroscopy*

FTIR-ATR spectroscopy was utilized to confirm the presence of characteristic chemical bonds in the fluorinated sol–gel silica materials (as powders). Figure 2 displays the FTIR-ATR spectra of fluorinated sol–gel silica materials at a wavenumber range of 4000–400 cm$^{-1}$. The important bands attributed to the characteristic vibrations of the network bonds vibrations are shown in Table 1.

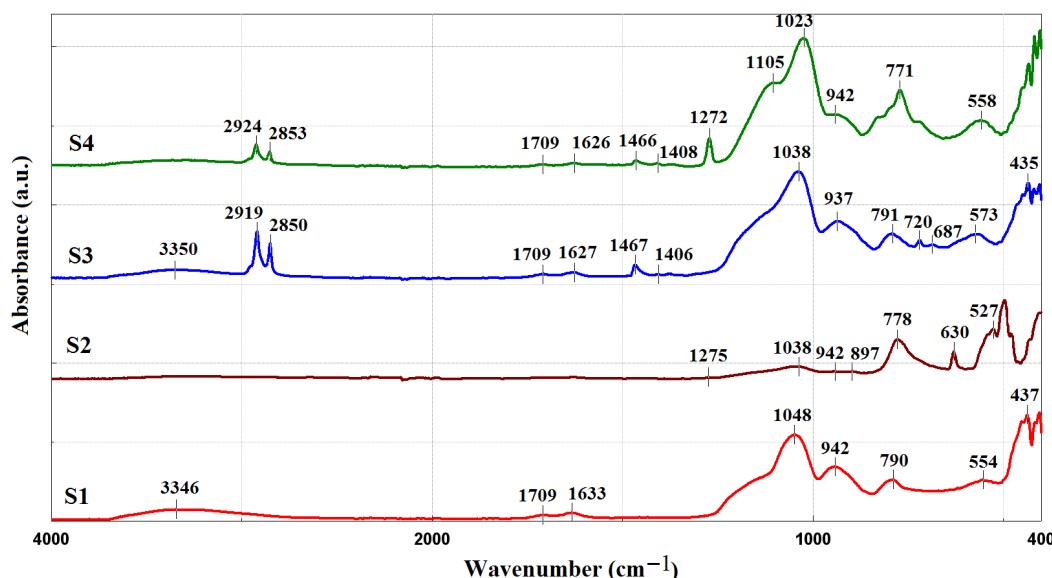

**Figure 2.** FTIR-ATR spectra of fluorinated sol–gel silica materials: TEOS+FS (S1), TEOS+MTES+FS (S2), TEOS+HDTMES+FS (S3), and TEOS+MTES+HDTMES+FS (S4).

**Table 1.** The assignment of the bands from the FTIR-ATR spectra.

| Wavenumbers (cm⁻¹) | Vibrational Modes |
|---|---|
| 3350 | O–H stretching |
| 2930–2850 | C–H stretching |
| 1633 | –OH bending |
| 1466 | C–H scissoring |
| 1272 | Si–C stretching |
| 1000–1100 | Si–O–Si stretching |
| 942 | Si–O stretching |
| 790 | Si–O–Si stretching |
| 771 | C–H out-of-plane |
| 558 | Si–O–Si |
| 520–730 | C–F rocking and wagging |

In the FTIR-ATR spectra of all fluorinated sol–gel silica materials, the bands specific for Si–O–Si stretching vibrations are identified in the range of 1020–1110 cm⁻¹ and correspond to the condensed silica network formation [24]. The band located at 942 cm⁻¹ is attributed to Si–O stretching in Si–OH in the xerogel silica materials. The peak at ~560 cm⁻¹ is assigned to symmetric stretching vibrations of Si–O–Si bonds, indicating that the linear silica networks were formed during the hydrolysis and condensation reactions [25,26]. A broad specific band is confirmed at ~3350 cm⁻¹ which can be related to the –OH group in H-bonded water (stretching vibration) [27].

In the FTIR spectra of samples, S1, S3, and S4, the peak at 1631 cm⁻¹ is identified and assigned to the –OH bending vibration [28].

Regarding samples S2 and S4, a peak at ~1272 cm⁻¹ is detected and corresponds to the Si–CH₃ stretching vibrations [29]. Additionally, in both samples, a peak of 770 cm⁻¹ is localized and attributed to the out-of-plane vibration of C–H bonds [30].

In the FTIR spectra of samples S3 and S4, other peaks at the 2925–2853 cm⁻¹ range are observed, characteristic of C–H bonds from –CH₃ and –CH₂– groups (stretching vibrations) present in aliphatic chains of the organosilane [31]. Another peak at ~1466 cm⁻¹ is observed in the FTIR spectra of these samples and is due to aliphatic CH– groups of the organic groups (scissoring vibration) [32].

The FTIR spectrum of sample S4 reveals a distinct additional new peak at 1100 cm⁻¹ as compared to other samples, which can be related to the stretching vibration of C–F bonds due to possible fluorination of silica materials by the PFOA molecules [33]. In previous studies, it was confirmed that the existence of C–F bonds are located in the range of 520–730 cm⁻¹ [34].

It is possible that the specific interactions (such as hydrophobic interaction, electrostatic forces, covalent bonding, hydrogen bonding, and van der Waal interactions) between the sol–gel silica network and the perfluorooctanoic acid (PFOA) were created during the condensation of the silica precursors (TEOS, MTES, HDTMES) in co-presence with the fluorinated solution (FS), thus explaining the differences in the bands related to the Si–O, C–H, Si–C, and C–F interactions of the fluorinated sol–gel silica materials.

It was reported that a high adsorption capacity for fluorinated compounds can be attained due to the hydrophobic interaction between the fluorinated chain and surface alkyl groups [35].

*3.2. Thermogravimetric Analysis (TGA)*

The TGA curves of fluorinated sol–gel silica materials are shown in Figure 3. Table 2 presents the decomposition temperature (T$_{max}$) and the weight loss (wt. loss %) of obtained materials.

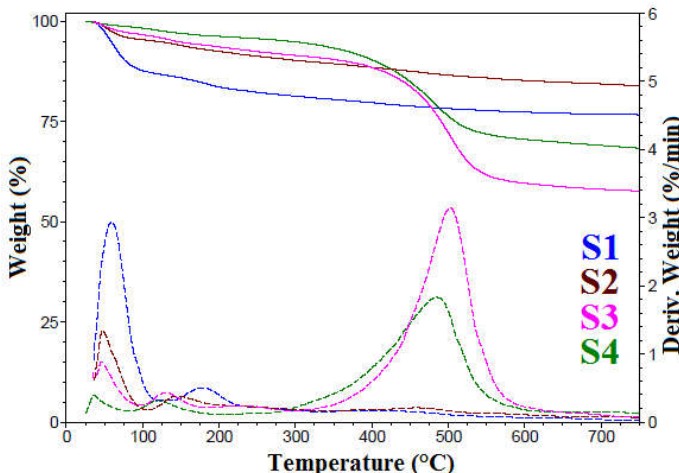

**Figure 3.** Thermogravimetric analysis curves of the fluorinated sol–gel silica materials (as powders): TEOS+FS (S1), TEOS+MTES+FS (S2), TEOS+HDTMES+FS (S3), and TEOS+MTES+HDTMES+FS (S4).

Figure 3 demonstrates that the process of weight change of the fluorinated sol–gel silica materials occurs in some sections. The thermal behavior of samples S1 and S2 are remarkably different from those of samples S3 and S4. In the first section of about 30–100 °C, the lost weight was attributed to the dehydration of the water and of residual organic solvent. In the second section at 100–340 °C, the lost weight was caused by the condensation process from Si–OH to form Si–O–Si, which released $H_2O$ as a byproduct, in addition to the decomposition of unreacted methyl and methoxy groups from silane precursors (MTES, HDTMES) [36]. The lost weight occurring at 340–750 °C was caused by the thermal decomposition of organic moieties (e.g., $–CH_3$ groups) functionalized on the silica surface.

**Table 2.** The decomposition temperature ($T_{max}$) and the weight loss (wt. loss %) of the fluorinated sol–gel silica materials, achieved as powders (temperature range of 30–750 °C).

| Sample | 30–100 °C | | 100–340 °C | | 340–750 °C | | Residue at 750 °C |
|---|---|---|---|---|---|---|---|
| | wt. Loss % | $T_{max}$ [1] °C | Wt. Loss % | $T_{max}$ °C | Wt. Loss % | $T_{max}$ °C | |
| S1 | 13.11 | 59.3 | 6.21 | 178.2 | 4.00 | 407.6 | 76.66 |
| S2 | 4.76 | 47.4 | 5.70 | 145.5 | 5.57 | 457.6 | 83.97 |
| S3 | 3.30 | 46.1 | 2.46 | 228.6 | 33.91 | 502.9 | 57.64 |
| S4 | 1.35 | 36.2 | 2.40 | 120.9 | 27.93 | 486.0 | 68.31 |

[1] $T_{max}$ (°C) = T $(d\alpha/dt)_{max}$.

Analyzing Table 2, it can be observed that the weight loss (below 750 °C) was 4.00 wt.% for sample S1, 5.57 wt.% for sample S2, 33.91 wt.% for sample S3, and 27.93 wt.% for sample S4. When dehydration is completed, dehydroxylation takes place, followed by the thermal degradation of samples between 340 and 750 °C. These results indicate that the presence of hydrophobic groups has a significant effect on the thermal degradation of the samples.

It was reported that, at high temperatures, oxidation occurs in the methyl group. These groups are responsible for the hydrophobic character of silica [37].

### 3.3. Ultraviolet/Visible Spectroscopy

In the existing literature, it has been shown that a surface with a micro- or nano-scale hierarchical roughness can be formed in connection with low surface energy [15].

The optical properties of the fluorinated hybrid films (conducted by the placement of fluorinated sol–gel silica materials on glass surfaces) are presented in Figure 4.

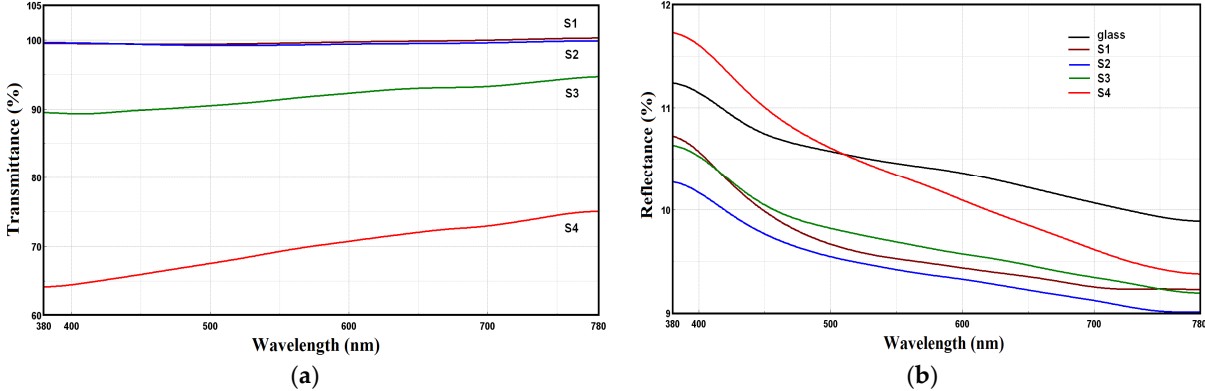

**Figure 4.** Transmittance (**a**) and reflectance (**b**) spectra of the glass surfaces: uncovered and covered with the fluorinated sol–gel silica materials: TEOS+FS (S1), TEOS+MTES+FS (S2), TEOS+HDTMES+FS (S3), and TEOS+MTES+HDTMES+FS (S4).

As shown in Figure 4a, the optical transmittance for the glass surfaces covered with fluorinated sol–gel silica materials varies from 69.5 to 99.0%, at 550 nm. It can be observed that the covered samples S1, S2, and S3 present a high optical transmittance. The fluorinated hybrid film S4 has low transmittance which is possibly due to the high refractive index of silica (n = 1.5 at $\lambda$ = 550 nm) [38]. The transmittance of covered samples S1 and S2 remained unchanged in the wavelength range of 600–780 nm compared with the coated samples S3 and S4. Analyzing the UV-Vis spectra of covered samples S3 and S4, it can be seen that the transmittance was changed, increasing (>92% and >70%, respectively) the wavelength range to 600–780 nm.

Analyzing Figure 4b, it can be observed that the reflectance of uncovered glass is about 10.5% at 550 nm. In the case of coated samples (S2–S4), the reflectance was ~9.5 at 550 nm. Analyzing the UV-Vis spectra of all coated samples, it can be observed that these coated samples exhibit a decreasing reflectance in the visible region.

It was demonstrated that the single porous silica layers allow a significant reflection reduction in the glasses in comparison with bare slide glass [39]. On the other hand, the antireflective properties of single-layer films depend on the destructive interference of the reflected light [40].

The clarity of films is a highly desirable characteristic for a variety of industrial applications (e.g., self-cleaning, solar panels, optical devices), as it indicates high quality and contributes to the progressive visibility of the material [41].

### 3.4. Atomic Force Microscopy (AFM) Measurements

Figure 5 shows the bi-dimensional (2D) AFM images recorded at the scale of (2 μm × 2 μm) for the samples S1–S4. Some morphological aspects can be noticed as follows. Samples S1 and S2 are relatively smooth, as their RMS roughness (denoted as Rq) values are below 1 nm: 0.44 nm (sample S1) and 0.574 (sample S2). Some random pits (or shallow cavities/valleys) are observed on both samples (see the dark spots/areas in Figures 5a and 5b). Similar to the roughness values are the so-called peak-to-valley (denoted as Rpv) parameters (representing the height difference between the lowest and the highest points on the scanned area), which are around 7–8 nm for both samples S1 and S2. On the other hand, in comparison with samples S1 and S2, samples S3 and S4 exhibit rough surfaces,

due to corrugated surface features (random crests), which lead to increased roughness parameters (see Figure 6): Rq = 8.38 nm/Rpv = 56.33 (sample S3) and, respectively, Rq = 4.04 nm/Rpv = 22.75 (sample S4).

In line with these observations, it is presumable that the roughness values will influence the contact angle values in the same order: S1 < S2 < S3 > S4.

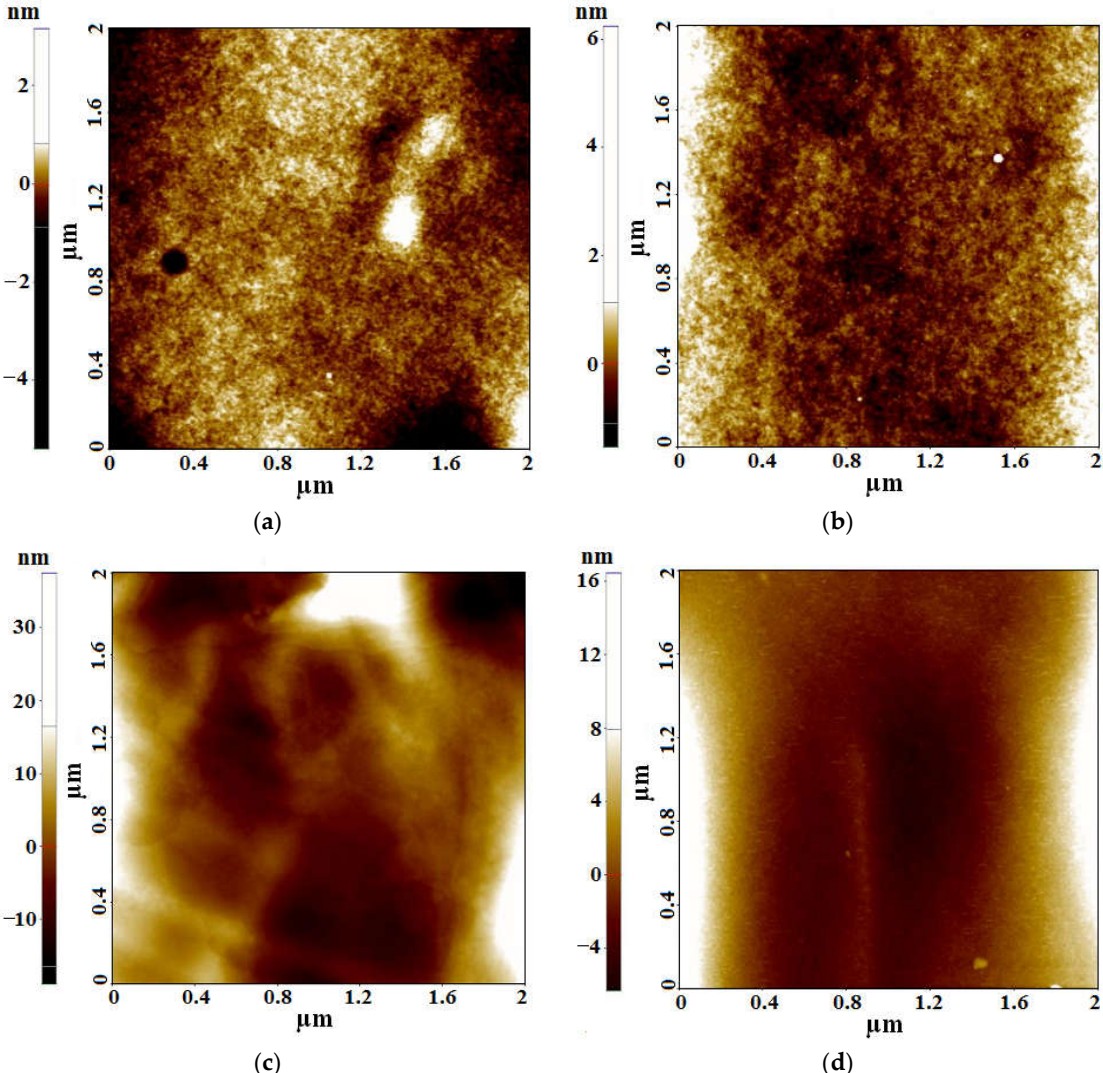

**Figure 5. The** 2D AFM images recorded at the scale of (2 μm × 2 μm) for glass surfaces covered with the fluorinated sol–gel silica materials: **(a)** TEOS+FS (S1), **(b)** TEOS+MTES+FS (S2), **(c)** TEOS+HDTMES+FS (S3), and **(d)** TEOS+MTES+HDTMES+FS (S4).

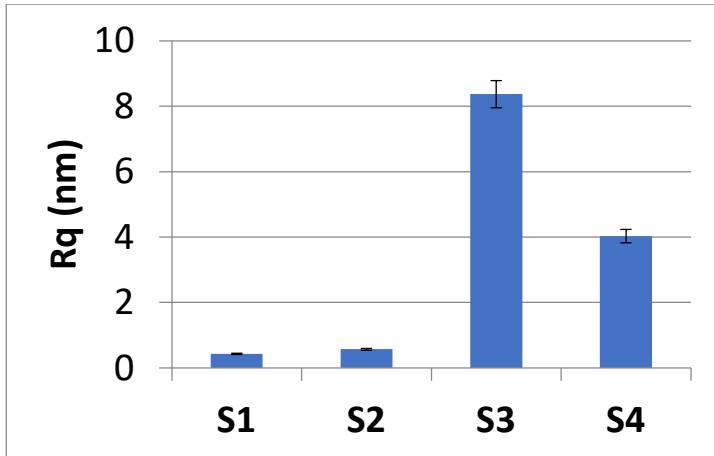

**Figure 6.** Root mean square roughness (Rq) at the scale of (2μm × 2μm) for glass surfaces covered with the fluorinated sol–gel silica materials: TEOS+FS (S1), TEOS+MTES+FS (S2), TEOS+HDTMES+FS (S3), and TEOS+MTES+HDTMES+FS (S4).

*3.5. Contact Angle Determinations*

The wettability of the uncovered glass surface and of glass surfaces covered with the fluorinated sol–gel silica materials was analyzed by static contact angle determinations. In Figure 7, the water contact angles values and profiles of water drops on the uncovered glass surface and covered glass surfaces are shown.

From Figure 7, it can be seen that the transition of the surface property was modified from a hydrophilic to a hydrophobic character. Modification of the surface led to an increase in the contact angle from 35 ± 1.6° for uncovered glass surfaces to 104 ± 1.5° for glass surfaces covered with the fluorinated sol–gel silica materials.

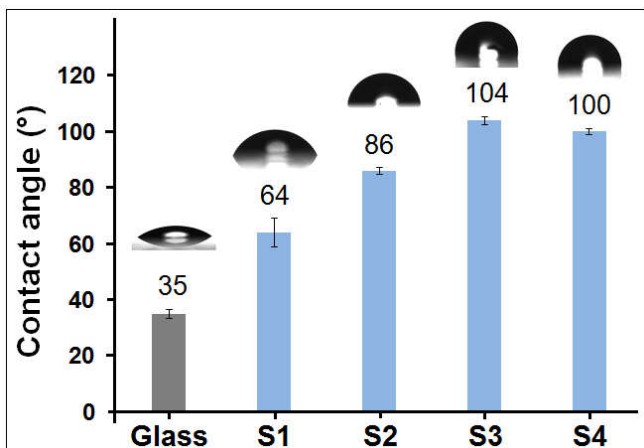

**Figure 7.** Water contact angle values and profiles of water drops on glass surfaces: uncovered and covered with the fluorinated sol–gel silica materials: TEOS+FS (S1), TEOS+MTES+FS (S2), TEOS+HDTMES+FS (S3), and TEOS+MTES+HDTMES+FS (S4).

The small water contact angles were identified for fluorinated hybrid films S1 and S2 (64° ± 5.1° and 86 ± 1.3°, respectively). It was demonstrated that, in a strongly acidic medium, the hydrolysis of the siloxane bridges (≡Si-O-Si≡ bond) conducts the formation of polar silanol (OH-Si≡ bond) groups at the surface. The hydroxyl groups are formed and oriented toward the polar water molecules. In this way, the hydrophilic character of the surface increased [42].

Fluorinated hybrid films S3 and S4 present the water contact angles of 104° ± 1.5° and 100° ± 1.1°, respectively, indicating that the hybrid surfaces were changed due to the silica matrix polarity. In this case, the silica becomes less hydrophilic after functionalizing the silane precursors because the partial hydrophilic sites were replaced by hydrophobic organic groups (hexadecyl). The water contact angle increases can be attributed to a large amount of air entrapment between the silica matrix and fluorinated segment forming a more ordered structure that diminishes the contact area of water from the surfaces [43].

Yang et al. [44] demonstrated the same trend in the modification of water contact angles of silica materials (using methyltriethoxysilane (MTES)).

In previous papers [45,46], it was shown that the values of the contact angle can be higher due to the segregation of the fluorinated segments on the surface and the high surface roughness of the coatings.

## 4. Conclusions

In this study, the fluorinated silica materials were successfully synthesized by sol–gel processing in an acidic medium using diverse silane precursors (TEOS, MTES, and/or HDTMES) and a fluorinated solution (FS). The fluorinated hybrid films were realized through the placement of fluorinated sol–gel silica materials on glass surfaces. The structure and surface properties of the final samples were analyzed via FTIR-ATR, ultraviolet/visible spectroscopy, TGA analysis, and water contact angle determinations. FTIR–ATR spectra indicated the differences in the bands related to the Si–O, C–H, Si–C, and C–F interactions of the fluorinated materials. TGA analysis of the fluorinated sol–gel silica materials showed that the dominant weight loss occurred in a temperature region between 350 and 500 °C. The fluorinated hybrid films present a high optical transmittance, which ranges from 91 to 99%. Ultraviolet/visible spectra revealed that these films had a reflectance of ~9.5%, at 550 nm. The fluorinated hybrid film covered with the sample that contained TEOS+MTES+HDTMES+FS presented a lower transmittance and higher reflectance compared to the other films, possibly because of the nanopores' generation on the functional material. The AFM results indicated that the samples prepared with HDTMES exhibited rough surfaces. Contact angle determinations indicated that the property of the surface can be modified from hydrophilic to hydrophobic. The fluorinated hybrid films covered with the samples that contain HDTMES (contact angle > 100°) present a hydrophobic character in comparison with the other samples. These fluorinated hybrid films can be useful in various electronics and optics fields.

**Author Contributions:** Conceptualization, V.P. and V.R.; methodology, V.P. and V.R.; formal analysis, F.M.R., A.R., A.N.F., M.A. and C.-A.N.; investigation, F.M.R. and A.R.; data curation A.N.F., M.A. and C.-A.N.; writing—original draft preparation, V.P., V.R. and S.C.; writing—review and editing, V.P., V.R., M.A. and S.C. All authors have read and agreed to the published version of the manuscript.

**Funding:** This research was funded by the INCDCP ICECHIM Bucharest 2019-2022 Core Program PN. 19.23–Chem-Ergent, Project No. 19.23.03.04.

**Institutional Review Board Statement:** Not applicable.

**Informed Consent Statement:** Not applicable.

**Data Availability Statement:** Not applicable.

**Conflicts of Interest:** The authors declare no conflicts of interest. The funders had no role in the design of the study; in the collection, analyses, or interpretation of data; in the writing of the manuscript, or in the decision to publish the results.

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
