# Peer review of "Investigation of Hybrid Films Based on Fluorinated Silica Materials Prepared by Sol–Gel Processing"

_coatings, doi:10.3390/coatings12101595_

Round 1
Reviewer 1 Report
This article requires a lot of precision. Regarding the synthesis, the role of aluminum alkoxide is quite vague and unexplained. The same is true for maleic acid. What is the mode of interaction of perfluorooctanoic acid with the silicate network? Sol-gel coatings are only dried at room temperature. What is the condensation rate of the materials obtained? A complete NMR analysis could have provided information on the silica network. The adhesion of the films could have been assessed. Regarding the thermal analysis, the authors claim to have worked in aluminum crucibles. Is this the right material for high temperatures? Moreover in figure 3, it would seem that the materials S2 and S4 have lower mass losses than those of materials with less organic...
It would be interesting to explain with experimental data the increase in contact angles for S3 and S4. If the authors suspect the presence of surface roughness why not measure it?
In general, it is absolutely important to increase the scientific quality of this paper, considering the fact that the sol-gel technology is now mature. The recipes in which all the arsenal of characterizations is not implemented are no longer admissible.
Author Response
Response:
Thank you for your special attention to this manuscript and for your suggestions. We appreciate all your comments.
- At section 2.1., it was mentioned the role of aluminum alkoxide and of maleic acid: “Aluminium isopropoxide (AIP, 98% purity, as reaction catalyst) was purchased from Sigma-Aldrich (Saint Louis, MO, USA). Maleic anhydride (MA, 99.7% purity, as complexing agent) was purchased from Fluka (Philadelphia, PA, USA).
- At section 2.2., new information was added: “Because the aluminium isopropoxide is very reactive, the addition of maleic anhydride as a complexing agent is necessary to control both the hydrolysis and condensation rate. Alkoxysilyl groups can actually co-condense during the condensation reaction involved in the sol-gel process, thus leading to the formation of the interaction between organic and inorganic species. Under acidic conditions, the hydrolysis rate is higher than con-densation rate of the materials obtained.”
- Regarding the thermal analysis, we worked in aluminum crucibles because is the right material for high temperatures.
- Table 2 present the weight loss (wt. loss %) and maximum decomposition temperature (Tmax) of obtained materials.
- At section 3.1., new information was added: “It was reported that, because of hydrophobic interaction between the fluorinated chain and surface alkyl groups, a high adsorption capacity for fluorinated compounds can be attained [1].”
- The increase in contact angles for S3 and S4 is explain in section 3.5: “Fluorinated hybrid films S3 and S4 present the water contact angles of 104° ± 1.5° and of 100° ± 1.1°, respectively, indicating that the hybrid surfaces were affected by the polarity of the silica matrix. In this case, the silica becomes less hydrophilic after functionalizing the silane precursors because the partial hydrophilic sites were replaced by hydrophobic organic groups (hexadecyl).”
- At section 3.4., it is presented the surface roughness (Figures 5 and 6): “Figure 5 shows the bi-dimensional (2D) AFM images recorded at the scale of (2µm x 2µm) for the samples S1-S4. Some morphological aspects can be noticed as follows. Samples S1 and S2 are relatively smooth, as their RMS roughness (denoted as Rq) values are below 1 nm: 0.44 nm (sample S1) and 0.574 (sample S2). Some random pits (or shal-low cavities / valleys) are observed on both samples (see the dark spots/areas in Figures 5a and 5b). In trend with roughness values are the so-called peak-to-valley (denoted as Rpv) parameters (which represents the height difference between the lowest and the highest points on the scanned area), which are around 7-8 nm for both samples S1 and S2. On the other hand, in comparison with samples S1 and S2, samples S3 and S4 exhibit rough surfaces, due to corrugated surface features (random crests), which lead to increased roughness parameters (see Figure 6): Rq = 8.38 nm / Rpv = 56.33 (sample S3) and, respectively, Rq = 4.04 nm / Rpv = 22.75 (sample S4).
In line with these observations, it is presumable that the roughness values will in-fluence the contact angle values in the same order: S1 < S2 < S3 > S4. “
- Unfortunately, other analyzes cannot be performed in Institutes and Universities from Romania because the indicated devices are broken and their repair requires an expensive financial effort. Please understand our views and support us in taking a favorable decision for this manuscript.

Reviewer 2 Report
In the present research, fluorinated silica materials were prepared by the sol-gel processing from tetraethylorthosilicate (TEOS), triethoxymethylsilane (MTES), trimethoxyhexadecylsilane (HDTMES), and using a fluorinated solution (FS), under acidic conditions. Some issuses should be addressed before publication:
(1) Page 2, Line 70-72, the authors say "In our previous studies, we have successfully demonstrated that the hydrophobic and antireflective films can be obtained by the sol-gel process, using the mixtures prepared with perfluoroalkylsilane (FAS13) and tetraethylorthosilicate (TEOS), at different molar ratios", could the author state, how the present work advanced the development on hydrophobic and antireflective films;
(2) Figure 4 could be improved to be prettier;
(3) Page 6, Line 195-198, "Is possible that the specific interactions between the silica sol-gel network and the perfluorooctanoic acid (PFOA) were generated during the condensation of the silica precursors (TEOS, MTES, HDTMES) in co-presence with the fluorinated solution (FS), thus explaining the differences in the bands related to Si–O, C–H, Si–C, and C–F interac-tions of the fluorinated sol-gel silica materials", could the author explain what's the "specific interactions between the silica sol-gel network and the perfluorooctanoic acid (PFOA)" in detail ?
(4) Page 7, Line 214-215, "At temperatures that are higher than this, further oxidation will occur on the methyl groups which are the groups responsible for the hydrophobicity of silica", for application scenarios of the prepared fluorinated silica materials, the temperatures that are higher than 500oC is not possible. So, the necessary for TG analysis is suggseted to be illustrated;
(5) Page 7, Line 238-239, " but the degree of reduction property depends on the precursor solution", the specific associations is suggseted to be explained in detail.
Overall the present work is funny and can be accepted after minor reversion.
Author Response
Response:
Thank you for your special attention to this manuscript and for your suggestions. We appreciate all your comments.
- In previous study, we used a much more expensive precursor perfluoroalkylsilane (FAS13) than the one used in this study (perfluorooctanoic acid (PFOA)). In this way, we obtained coatings using less expensive materials.
- Figure 4 was improved.
- At section 3.1., the specific interactions were mentioned: “such as hydrophobic interaction, electro-static forces, covalent bonding, hydrogen bonding and van der Waal interactions”.
New information was added: “It was reported that, because of hydrophobic interaction between the fluorinated chain and surface alkyl groups, a high adsorption capacity for fluorinated compounds can be attained [35].”
- At section 3.2. (TGA) new information was added.
Table 2 present the weight loss (wt. loss %) and maximum decomposition temperature (Tmax) of obtained materials.
“Analyzing Table 2, it can be observed that the weight loss (below 750°C) was 4.00 wt.% for sample S1, 5.57 wt.% for sample S2, 33.91 wt.% for sample S3, and 27.93 wt.% for sample S4. When dehydration is completed, dehydroxylation takes place, followed by the thermal degradation of samples between 340 and 750 °C. These results indicate that the presence of hydrophobic groups has a significant effect on the thermal degradation of the samples.”
- This phrase was corrected. New information was added: “On the other hand, the antireflective properties of single-layer films depend on the destructive interference of the reflected light [40].”

Round 2
Reviewer 1 Report
The paper can be published as is even if the answers to all of my questions are not present. However, the scientific quality is better.